# Digital Workflow for Implant Placement and Immediate Chairside Provisionalization of a Novel Implant System without Abutment—A Case Report

**DOI:** 10.3390/medicina58111612

**Published:** 2022-11-08

**Authors:** Felicitas Hölken, Bilal Al-Nawas, Manuela Meereis, Monika Bjelopavlovic

**Affiliations:** 1Department of Prosthodontics and Materials Science, University Medical Center of University of Mainz, Augustusplatz 2, 55131 Mainz, Germany; 2Department for Oral and Maxillofacial Surgery, Plastic Surgery, University Medical Center of University of Mainz, Augustusplatz 2, 55131 Mainz, Germany; 3Department of Prosthodontics and Materials Science, Dental Laboratory, University Medical Center of University of Mainz, Augustusplatz 2, 55131 Mainz, Germany

**Keywords:** implant surgery, implant-prosthetic interface, CAD/CAM, screw-retained, no abutment, digital

## Abstract

Dental implants have been placed millions of times worldwide, and the surgical procedure and implant design have steadily improved. The basic prosthetic connection, which makes use of an abutment, has changed little over the past decades. These days, implant placement with immediate provisionalization is an essential stage in implant dentistry and interdisciplinary treatment strategies. Temporary computer-aided design and computer-aided manufacturing (CAD/CAM) of implant-supported crowns restore esthetics and guarantee function during the restoration process and the osseointegration of the dental implant. This case report describes the digital planning; the immediate, static, computer-assisted implant surgery, and the immediate chairside provisionalization of a novel implant system that is directly screw-retained without an abutment.

## 1. Introduction

The CAD/CAM (computer-aided design/computer-aided manufacturing) systems used for decades in restorative dentistry have expanded their applications to implant dentistry. Today, these systems enrich the dental implant treatment with the diagnosis and planning as well as in the actual surgical and prosthetic procedures [1,2]. With continuous improvements in intraoral scanning technology, material innovations, and new software updates, this technique is becoming increasingly prevalent [3,4,5,6]. However, the implant interfaces have remained unchanged for the past decades. The innovation of the novel implant system (Matrix Implant, TRI Swiss Implants, Hünenberg, Switzerland) is the implant-prosthetic connection: due to CAM processing, no abutment is needed for the prosthetic construction, and the restoration is directly connected with the implant. Single-unit restorations with a corresponding implant-prosthetic connection can be milled in one piece from prefabricated temporary blocks or zirconiumoxide for definitive solutions. The bonding to a titanium base is, therefore, no longer necessary. According to the manufacturer, there is no limitation in angulation (up to 100° between implants), and it allows for direct restorations from screw-retained, fully anatomic CAD/CAM crowns to multi-unit bars and bridges, which can be planned and placed directly on the implant. The digital planning enables a patient-specific, individual emergence profile. The pink anoidized neck optimizes translucency in the gingival tissue. The geometry of the interface in the shape of an open circle can be produced with any kind of suitable CAD/CAM material without the necessity of an opening of a prefabricated protection against rotation. In addition to the conventional impression taking, the system is approved for a fully digital workflow, and the restorations are CAM-milled on a five-axis milling unit or 3D printed. The direct screw-retained connection of the restoration to the implant eliminates the need for cementing the restoration or bonding to a TiBase; consequently, a potential cause of peri-implantitis can be prevented. Although cemented restorations may have an esthetic and clinical advantage, cement residues are often associated with biological implant pathologies [7]. In particular, they can provide a retention zone for plaque and subsequently lead to inflammation signs, which can quickly develop into peri-implantitis and result in large losses of soft and hard peri-implant tissues [7]. The use of CBCT allows bone density to be assessed and the expected bone contact with the implant to be visualized [8]. The preoperative planning for the implant position or surgical templates is further improved through virtual planning tools and digital workflows. In addition, the treatment gains predictability and reduces surgical morbidity [9]. To minimize the risk of implant failure, the dentist can decide whether prosthetic immediate loading is possible or whether a delayed restoration is preferred [8]. Nowadays, the patient appears for implant placement and is directly discharged chairside with a CAD/CAM-milled restoration or a restoration that has been digitally planned and fabricated in advance.

Implant placement with immediate provisionalization is an essential stage in implant dentistry and interdisciplinary treatment strategies. Immediate interim prostheses can be used in situations when the bone volume is ideal, there is no guided bone regeneration procedure, and there is good primary stability [10,11,12]. After the presence and shaping of soft tissue, the temporary restoration helps form the emergence profile [13,14,15,16,17,18,19]. The dental team can evaluate the situation with regard to its esthetics, phonetics, and masticatory function [20]. According to the fabrication technique, provisional restorations are divided into direct, indirect-direct, and indirect restorations [18]. The direct technique is made intraorally with autopolymerizing materials and without the need for an implant level impression [21]. Due to the conditions under which they are polymerized and fabricated, these restorations are susceptible to pores, cracks, and inhomogeneities, which may lead to bacterial contamination, discoloration, bacterial ingress, and a significant decrease in long-term stability and biocompatibility [22]. In the indirect technique, the prosthesis is constructed in the laboratory and is more color stable, less porous, more wear resistant, and esthetically enhanced as compared to direct restorations [21]. CAD/CAM is a promising technology that has completely changed the prosthetic manufacturing [23,24]. The advance of this science and a high demand for metal-free restorations have led to rapid advancements in the development of newer restorative materials and processing technologies [25]. Its use, digitally generated datasets, and numerical control (NC) allows us to work with new, industrially prefabricated, and almost defect-free restorative materials [26]. While, in the past, almost exclusively manual techniques were used to produce restorations, CAD/CAM technology now helps us to process completely new materials, which are unfit for manual processing. Since they are manufactured in an industrial process, provisional restorations made of high-density polymer (polymethylmethacrylate (PMMA)-based high-density polymer) or composite-based polymer blanks exhibit qualities superior to those of direct temporary restorations [27,28,29,30], with increased long-term stability, biocompatibility, and resistance to wear [18]. According to the latest findings by Reda et al., BioHPP is a promising material for immediate provisionalization and loading due to its mechanical properties and biocompatibility [31].

The objective of this case report is to describe the treatment of a digital workflow for implant placement and direct chairside, screw-retained provisionalization of a novel implant system. 

## 2. Case Report

A 34-year-old female patient presented to the Prosthodontic Department of the Dental Clinic of the University Mainz (Table 1).

Tooth 14 had been extracted 4 years ago, and teeth 13 and 15 had already been prepared for an FDP. The patient did not accept the foreign body sensation of the temporary bridge pontic and requested implantological treatment (Figure 1).

A radiological image of tooth 15 showed an insufficient root canal treatment with apical extrusion of the root canal filling and an apical translucency, PAI = 3. The patient was referred for endodontic revision treatment. Teeth 13 and 15 received new long-term temporaries (LTT) (Telio CAD, Ivoclar Vivadent, Schaan, Liechtenstein). The bone height was sufficient (measured on a CBCT scan). Suitable anatomical conditions (mesial-distal, buccal-palatal, and interocclusal space) to place an anatomically designed screw-retained restoration were present and complied with good oral hygiene practices. The patient had no medical and general contraindications for the surgical procedure and did not smoke or show severe bruxism with dysfunctional tendencies. The presence of acute untreated periodontitis in the implant bed or adjacent tissue could be excluded. Written informed consent was obtained from the patient before being enrolled for implant treatment. The patient underwent implant placement and chairside provisionalization according to the treatment procedures described in Table 2.

## 3. Diagnostic and Planning Procedures

Prior to the implant treatment, intraoral scans of the upper jaw, lower jaw, and a lateral bite registration (CEREC Primescan SW 5.2. Dentsply Sirona, Charlotte, USA) as well as a cone beam computed tomography (CBCT) scan were made (Figure 2). 

The intraoral scans and CBCT Digital Imaging and Communications in Medicine (DICOM) file were superimposed with an implant planning software (coDiagnostics SW 10.6, Dental Wings, Chemnitz, Germany) to create an individualized digital set-up (Figure 3).

The prosthetically driven implant position was decided by determining the tooth axis and the natural bone width in a palatal position with a ≥2 mm distance between the implant and the buccal crest and a depth of 3–4 mm apical of the prospective restorative zenith point [32]. The implant length was chosen based on the available bone height. A pilot-drill surgical template to facilitate static, computer-assisted implant surgery was designed with the implant planning software and printed (SolFlex, W2P; Austria) (Figure 4).

## 4. Surgical Procedures

The surgical procedure was performed using local anesthesia (Ultracain D-S forte, Sanofi-Aventis Deutschland GmbH, Frankfurt am Main, Germany). A marginal incision with mesial and distal relief to expose the bone and prepare a minimal mucoperiosteal flap was made in order to obtain an intraoperative overview and to assess the bone quality (Figure 5). 

In order to gain proper primary stability, a pilot drill (Tri Swiss Implants, Hünenberg, Switzerland) with the use of a surgical template was utilized (Figure 6). 

A second drill (Tri Swiss Implants, Hünenberg, Switzerland) was used to prepare the coronal third of the alveolus. The neck area was prepared with a countersink drill. A tissue-level tapered implant (Matrix Tissue-Level Implant, TRI Swiss Implants, Hünenberg, Switzerland) with a Ø 3.3 mm enossal (Ø 3.7 mm platform) and a length of 11.5 mm was inserted according to the manufacturer’s instructions (Figure 7, Figure 8 and Figure 9). 

Primary implant stability was attained with a maximum insertion torque of 45 Ncm, verified with a manual torque controller to check if the implant stability was sufficient for immediate temporalization. The scanbody was screwed onto the implant and closed using sutures (5-0 polytetrafluoroethylene) to complete the treatment. The surgical procedures were performed by one oral and maxillofacial surgeon.

## 5. Restorative Procedures

The implant situation was digitalized using a scanbody (TRI Swiss Implants, Hünenberg, Switzerland; Cerec Primescan, SW 5.2., Charlotte, USA) (Figure 10 and Figure 11). 

The screw-retained temporary crown was designed with a CAD/CAM screw channel using a dental CAD Software (exocad, SW 3.0 Galway, Darmstadt, Germany). The geometry had been stored in the CAD Software catalog with default settings for PMMA-based, high-density polymer blocks (Telio CAD, Ivoclar vivadent, Schaan, Liechtenstein) (Figure 12) and fabricated utilizing a five-axis milling unit (imes icore 350i, Eiterfeld, Germany). Consequently, materials without a prefabricated screw channel can be used.

The final restoration was tightened onto the implant with a torque value of 15 Ncm using a manual torque controller (Figure 13). The screw access hole was sealed with polytetrafluoroethylene (PTFE) tape and composite resin. The restoration was free from centric and eccentric movements, and the patient was instructed to avoid excess force during the healing period. Oral hygiene instructions were given. The dental implant’s position was checked using X-ray (Figure 14). All the laboratory procedures were carried out by one dental laboratory. The restorative procedures were performed by one dentist. 

## 6. Discussion

Implant treatment planning has changed in recent years from a surgical approach that focused mainly on bone availability to prosthetic-oriented planning with 3D imaging and CAD software. CBCT has ushered in a new era of diagnostic capabilities in the dental office. Nowadays, a major advantage of implant surgery is the computer-guided, template-assisted surgery, especially in complex situations. Better clinical experiences and evidence-based improved outcomes can be offered to patients with confidence when CT-guided dental implant surgery is used [33,34].

The implant placement and the immediate provisionalization require accurate treatment planning. The preoperative digital planning aided in choosing an implant length and diameter that fit within the biologic boundaries position [35,36] and was based on the prosthetic requirements. Assessing the three-dimensional alignment and depth of the first drill, using a pilot-drill surgical template, are crucial steps in the preparation of the implant site. This approach is very helpful for single implants with narrow conditions, as the diameter of the sleeve is reduced, and it can be positioned between two adjacent teeth. The static, computer-assisted immediate implant surgery with the pilot-drill surgical template allowed the clinician to perform the rest of the procedure using freehand techniques and resulted in the present case of a correct three-dimensional implant placement. It must be discussed that a fully guided surgical technique can offer more accuracy and that a certain degree of deviation from the digital planning should be taken into consideration when using a pilot-drill surgical template [37,38]. Primary stability of the implant was reached, which allowed immediate provisionalization in the same appointment. 

The aims of fixed immediate prostheses include patient comfort, proper management of soft tissue, and elimination of second-stage surgery [39]. Especially in the field of esthetic surgery, such as in the anterior or premolar region, the immediate temporary restoration also offers advantages from an esthetic point of view. The emergence profile is optimally formed by conditioning the peri-implant tissue, and papillae are, thus, formed at an early stage; a clear line between the subgingival and supragingival areas can be seen [40]. The restorations of the novel implant system directly planned on the implant (without abutment) also enable an individual emergence profile for definitive solutions. Although almost all dental implant systems have prefabricated provisional abutments for the fabrication of the fixed interim, these components are time-consuming and costly to handle manually. 

In the past, implants were restored using metal abutments and porcelain fused to metal crowns. The use of metal-ceramic components leads to higher material and processing costs. Veneered crowns do not achieve the esthetic perfection of all-ceramic crowns. The trend towards materials with improved biomimetic properties, such as all-ceramic restorations, has enabled us to deliver better esthetic results in terms of the appearance of our restorations [41,42,43,44,45,46,47,48]. Even though modern restoration modalities promise a fully digital workflow, such as scanned implant situations and CAM-fabricated restorations, the latter still needs to be manually cemented or bonded. Today, titanium bases bonded on CAM ceramic abutments or crowns are frequently used. Prefabricated titanium base abutments are adhesively connected to a ceramic abutment (=hybrid abutment) or to a full contour crown (=hybrid abutment crown) [49]. Due to these hybrid abutments, deep submucosal abutment shoulders are avoided and, thus, also excess cementum, which can lead to inflammatory reactions of the gingival tissue with adjacent marginal bone loss [50,51,52].

Although titanium and ceramics are known for their high biocompatibility, the adhesive luting agent may cause irritation, is susceptible to aging [53], and may develop defects at the interface, which, in turn, are susceptible to bacterial colonization [54]. Unpolymerized monomer of the bonding material may have a negative effect on the peri-implant tissue health.

In this context, the idea of a directly screwed interface of the novel implant without adhesive luting seems tempting. However, it should be considered that the direct connection between the implant and the build-up shows always a microgap, even if they are connected correctly [55]. The gap can enlarge over time due to masticatory loading, which can lead to micromovements of the prosthetic components [54,55,56,57]. The microleakage caused by the microgap allows the passage of acids, enzymes, and bacteria [58]. The accumulation of microorganisms around the implant can cause soft tissue infections and bone loss around the implant, which can lead to implant failure [54,56,59].

The fabrication of the provisional restoration from tooth-colored composite or PMMA materials or the definitive restoration with zirconiumoxide of the novel implant system promise an esthetically pleasing restoration. The pink anodized implant surface has a positive effect on gingival esthetics and prevents dark margins. It masks any flashing through the mucosa, which can be confirmed in the present case. The concave design of the implant shoulder may have a positive effect on the surrounding soft tissue. The tapered implant design is suitable for immediate implant placement. As in the present case, it is well designed for narrow gaps with root proximity of the adjacent teeth to avoid injuring anatomical structures [60]. Due to its screw design, it offers a high primary stability even in soft bone situations (D4) as it condenses bone locally, resulting in more implant stability [61,62].

CAD/CAM materials are increasingly used as temporary implant support or as LTT restorations [63,64,65]. In this case report, we used high-density polymers (Telio CAD, Ivoclar Vivadent, Schaan, Liechtenstein) for LTT restorations. According to the manufacturer, the material is approved from single-tooth restorations up to 4-unit FDPs, including restorations on implants. High-density polymers, fabricated under industrial conditions to form a highly homogeneous structure [18], offer more wear resistance [66] and stability and biocompatibility over conventional composite resin or acrylic materials [18,30,67,68,69,70].

Comparing conventional and digital prosthetic working methods, a reduction in working time can be seen [71]. The intraoral scanning lowers the patient discomfort and procedure time compared to conventional impression taking [72]. Furthermore, the CAM process can reduce the production time and fabrication costs of the restorations [73]. The absence of the abutment of the novel implant system eliminates the step of bonding the restoration in the laboratory and further shortens the working time by using internal computerized techniques even more. In the long run, the supply will be cheaper due to fewer materials and work steps. However, it has to be considered that initial operating expenses are very high. The financial investments for intraoral scanners, CAD/CAM software, license fees, and the milling unit are high in comparison with conventional workflows [5]. 

In the past, the literature showed that the geometry of the abutment and the presence of a screw channel are critical factors for the success of a fixed dental restoration [74,75,76,77].

Chewing forces might be directly transferred from the crown to the abutment, leading to a higher stress on the screw and the implant-abutment interface [78], whereas another study, examining restorations with Ti-Base abutments, showed that the presence of a screw channel had no clinically relevant impact on the stability of the crown. In this context, the author suspects that the Ti-Base abutment with comparatively sharper edges, a smaller diameter, protection against rotation, and a steeper preparation angle is responsible for this [50]. Therefore, it remains to be seen how the novel implant system behaves in the absence of an abutment and with a direct screw-retained connection between the restoration and the implant.

A limitation of the case report is the narrow diameter of the implant. Narrow diameter implants provide a reduced surface area for osseointegration; they may be at increased risk of overload due to occlusal forces [79,80]. Choosing a 3.3 mm enossal diameter (and 3.7 mm platform diameter) of the implant in the present case should be mentioned. The narrow diameter has been proven to be useful in challenging indications where the width of the alveolar ridge or the interdental space is insufficient. The implant provides more space for the bone and soft tissue in tight conditions [79,81,82]. However, the distance to the adjacent teeth was maintained in this case.

An alternative treatment option for the patient could have been a removable prosthesis. In view of the patient’s age, a fixed restoration was agreed upon. The shortened dentition is to be restored at a later date by another implant placement with bone augmentation measures in the molar region. The outcome of the revision treatment of tooth 15 remains to be seen.

Nevertheless, it is a limitation of this novel implant system that data from clinical studies are missing. 

## 7. Conclusions

The single-unit restorations of the novel implant system can be milled under additional polishing (temporary restorations, e.g., PMMA) or crystallization (final restoration with zirkoniumdioxide) without a time-consuming mechanical attachment to a titanium base. The absence of the cement may have a positive effect on the surrounding tissue and healing. Due to its tooth-colored restoration and the pink anoidized implant surface, they are interesting for highly esthetic situations with fully digital treatment paths. Results from additional clinical studies are required to validate the positive results from these initial clinical experiences.

## Figures and Tables

**Figure 1 medicina-58-01612-f001:**
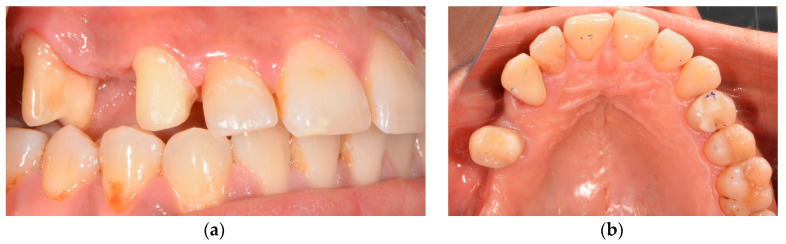
Initial situation: (**a**) lateral view: tooth 15 and 13 already prepared and tooth 14 missing for several years; (**b**) occlusal view: temporary restorations placed on teeth 15 and 13.

**Figure 2 medicina-58-01612-f002:**
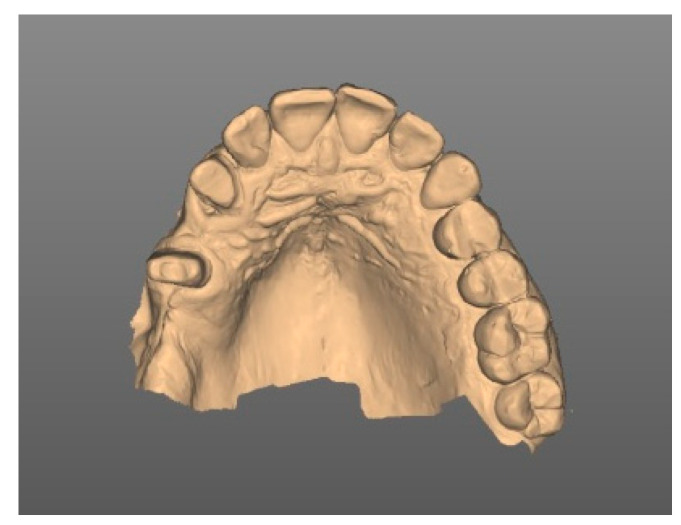
Digital impression of the upper jaw.

**Figure 3 medicina-58-01612-f003:**
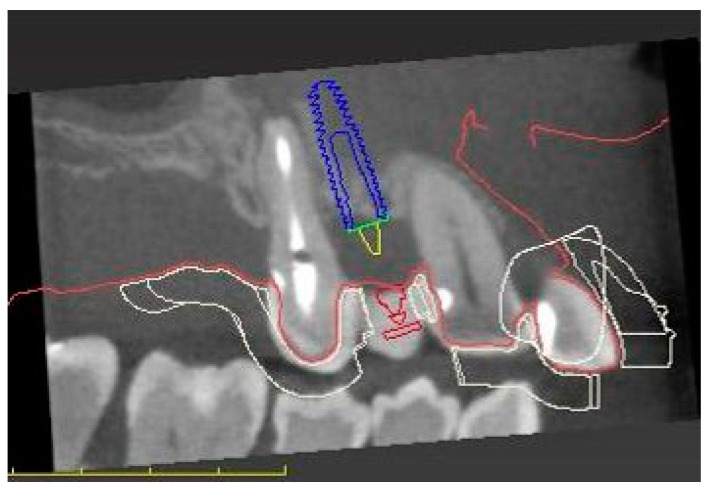
Digital planning of the implant and surgical template position.

**Figure 4 medicina-58-01612-f004:**
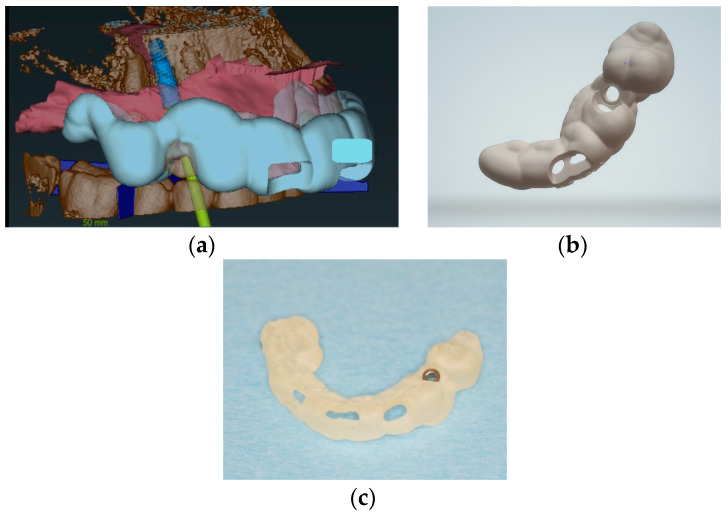
(**a**) Digital design for implant position and surgical template, coDiagnostics; (**b**) digital design of the surgical template; (**c**) surgical template with a sleeve in region 14.

**Figure 5 medicina-58-01612-f005:**
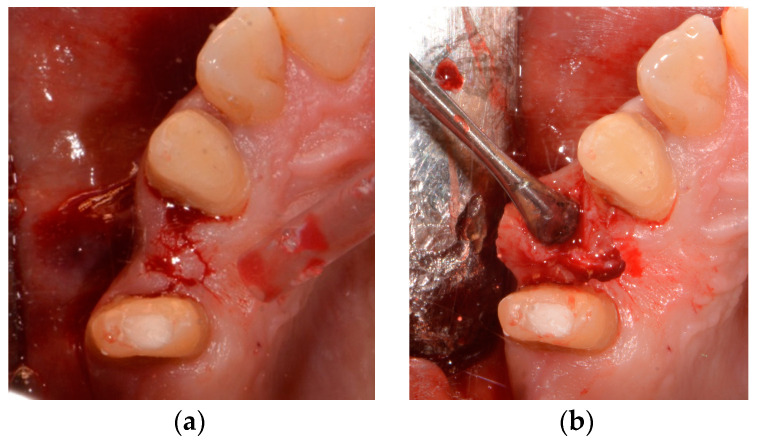
(**a**) Initial cut in region 14; (**b**) raising of the minimal mucoperiosteal flap in region 14.

**Figure 6 medicina-58-01612-f006:**
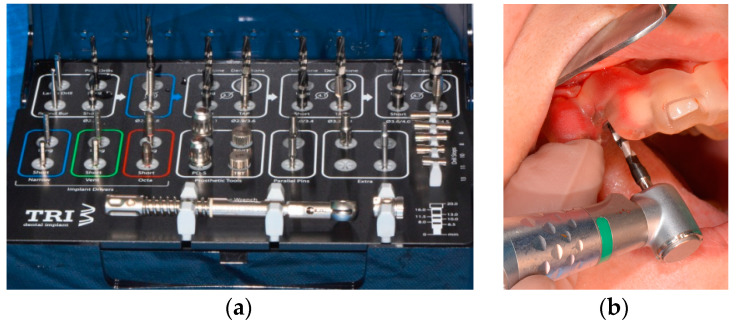
(**a**) Surgical cassette TRI dental implants; (**b**) implant bed preparation with tapered drills using a surgical template.

**Figure 7 medicina-58-01612-f007:**
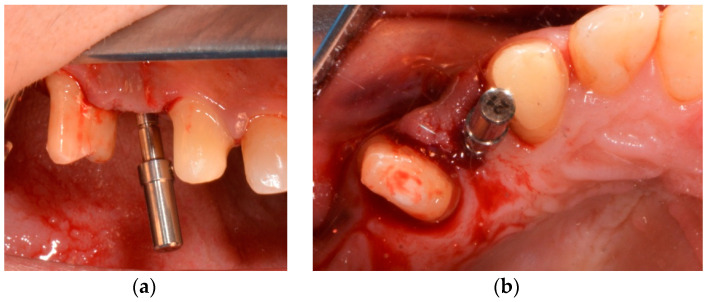
(**a**) Control of the position with the implant direction indicator; (b) control of the position with the implant direction indicator.

**Figure 8 medicina-58-01612-f008:**
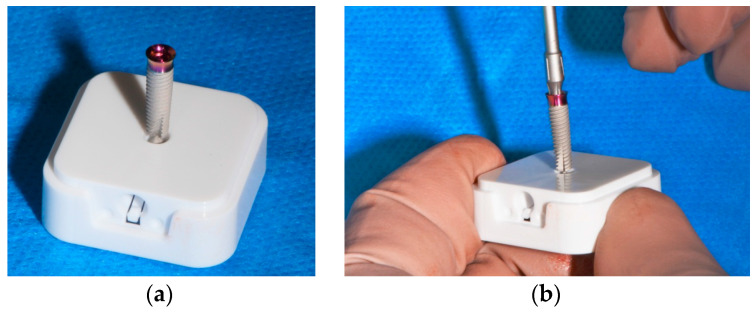
(**a**) Matrix Tissue level, TRI Implant Swiss; (**b**) picking up the implant.

**Figure 9 medicina-58-01612-f009:**
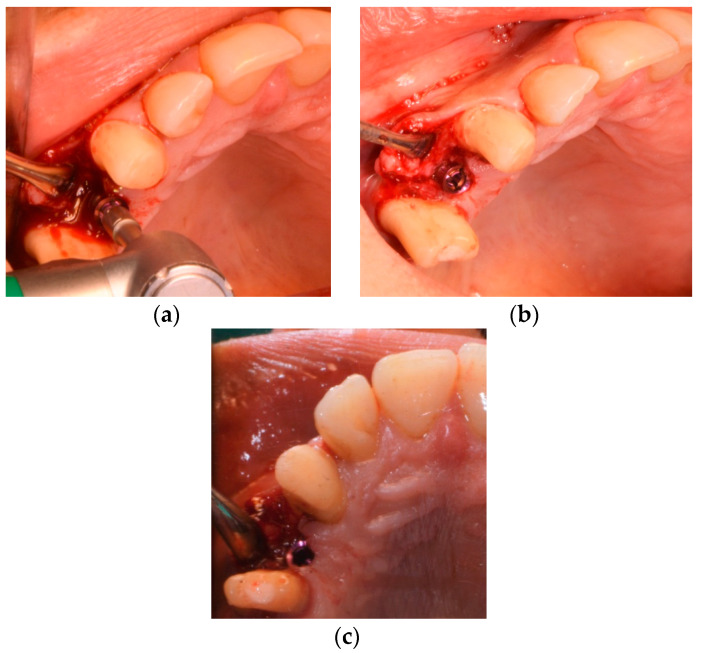
(**a**–**c**) Implant insertion.

**Figure 10 medicina-58-01612-f010:**
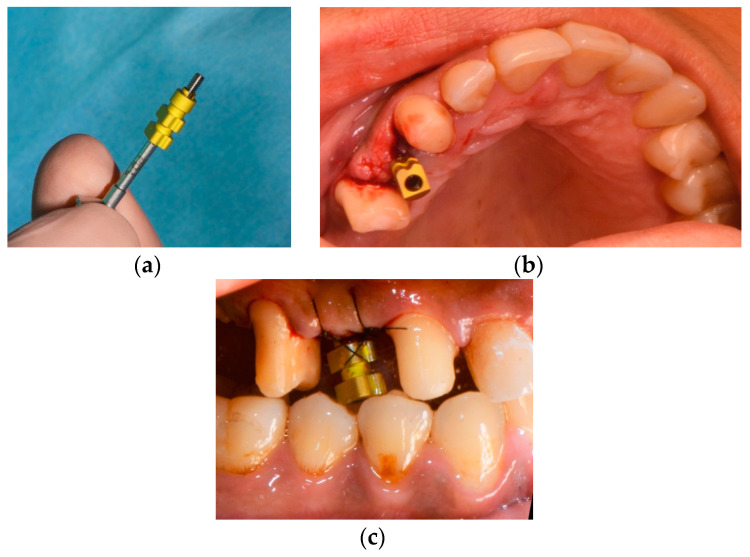
(**a**) Picking up the ScanPost; (**b**) inserted ScanPost; (**c**) inserted SscanPost with an adapted single suture for repositioning the mucoperiosteal flap.

**Figure 11 medicina-58-01612-f011:**
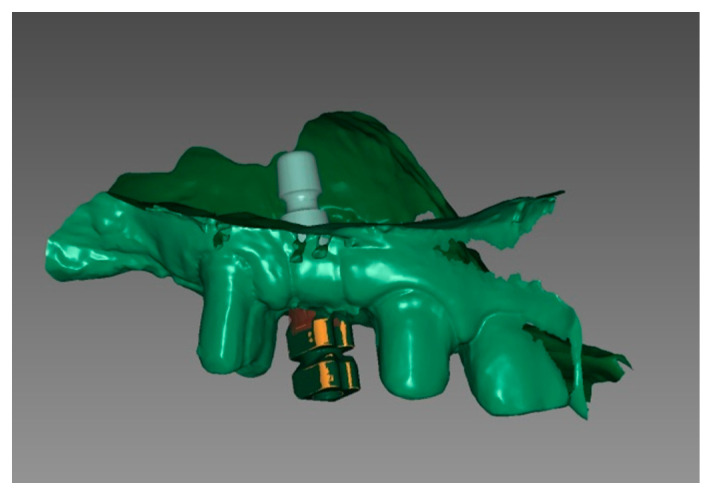
Chairside digital impression in region 14, exocad, SW 3.0 Galway.

**Figure 12 medicina-58-01612-f012:**
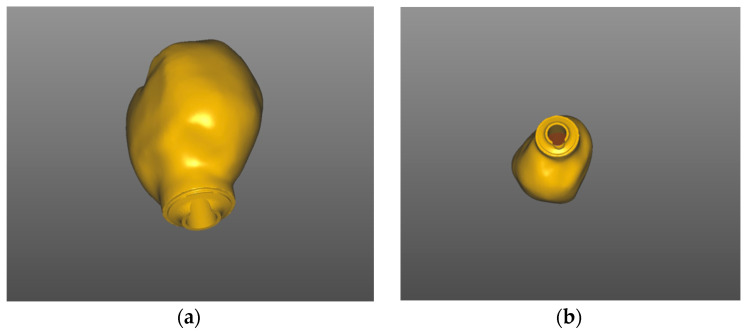
Chairside digital design for implant 14 exocad, SW 3.0 Galway: (**a**) bucco-distal view; (**b**) bucco-inferior view.

**Figure 13 medicina-58-01612-f013:**
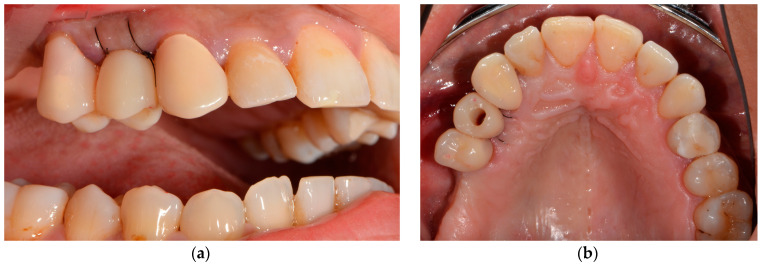
(**a**) Custom-made, temporary chairside restoration of implant 14; lateral view; (**b**) custom-made, temporary chairside restoration of implant 14; occlusal view.

**Figure 14 medicina-58-01612-f014:**
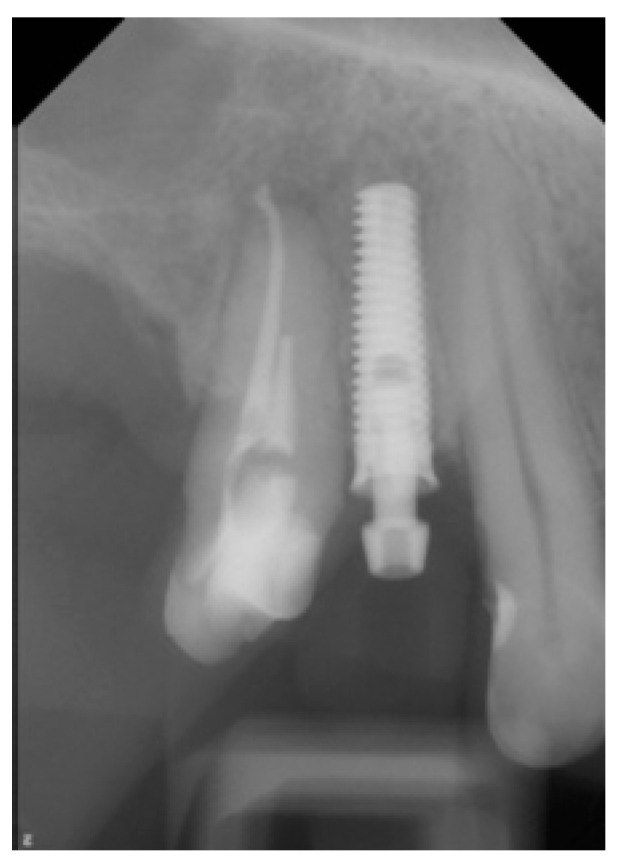
Dental X-ray after dental implantation.

**Table 1 medicina-58-01612-t001:** Patient information.

Treatment Phase	Procedures
Gender	female
Age	32
Implant position	Tooth 14 (upper left first premolar)
Implant type	Tissue-Level (Matrix, TRI Swiss Implants)
Implant diameter platform	3.7 mm
Implant diameter enossal	3.3 mm
Implant length	11.5 mm

**Table 2 medicina-58-01612-t002:** Timeline of the diagnosis, planning, implant placement, and chairside provisionalization.

Treatment Phase	Procedures
Diagnostic	Intraoral scans
	Cone beam computed tomography scan
Planning	Digital set-up
	Digital prosthetically driven implant planning
	Manufacturing of the surgical template
Surgical	Computer-assisted implant surgery
Restorative	Intraoral scans with a scanbody
	Chairside, computer-aided design and milling of the provisionalization
	Placement of the chairside LTT

## Data Availability

The authors confirm that the data supporting the findings of this study are available within the article.

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
