# Peer review of "Digital Workflow for Implant Placement and Immediate Chairside Provisionalization of a Novel Implant System without Abutment—A Case Report"

_medicina, 2022, doi:10.3390/medicina58111612_

Round 1

Reviewer 1 Report

Dear Authors, 

you made a great work! However, some improvements are mandatory before acceptance. 

Author Response

Dear Reviewer 1,

We wanted to thank you for your encouraging and detailed review. Here are our changes and answers to your comments in a point-to-point reply.

“The temporary restoration can be useful for controlling the osseointegration (12) and helps forming the emergence profile (13-19).” How is it useful to check integration? and regarding the emergence profile, I would emphasize more than anything else the importance of soft tissue support, primary compared to the emergent profile of the restoration.

-> Thank you for your aspects, we managed to emphasize the influence of the soft tissue for the emergence profile and we deleted the part that the temporary restoration can be useful for controlling the osseointegration. Line 52,53.

“Single unit restorations can be milled from prefabricated temporary blocks or zirconiumoxide for definitive solutions” do you mean to link on a titanium base or directly by milling the implant connection in zirconia?

-> Thank you for this aspect. The implant prosthetic connection is directly milled, even in ziconiumoxide.  The bonding to a TiBase is not necessary anymore. We have recorded this in line 44-47.

“The direct screw retained connection of the restoration to the implant eliminates the need for cementing the restoration or bonding to a TiBase: consequently, a potential cause of peri-implantitis can be prevented.” I ask the authors to enrich and better explain these passages, fundamental in the introduction and key concept of the article.

-> Thank you for your aspect. The implant prosthetic connection is the key factor of this article. We emphasized this part in line 55-61.

As a Clinician I confess to you that I have been using monobloc and ceramic coated Cr-Co solutions for years, when the implant axis allows me to use it on a single restoration. Therefore I am very attentive to this article, please explain better the materials used, and the digital techniques also in the introduction. What you have correctly explained in the introduction, you can also find it here and also regarding the use of materials with high functional value for immediate provisionalization or loading:

-> Thanks for the feedback and the suggested articles, a very interesting aspect that I have taken up and considered in the text. According to the latest findings of Reda et al. BioHPP should be mentioned for the immediat supply. line 41-57 and 76-96

please check the layout of the manuscript.

-> Thank you for pointing this out. The layout was processed.

In the case description:

 “t (Error! Reference source not found.).” Please check.

->Thank you for this hint, we managed to link every figure, table reference to a source.

“A marginal incision with mesial and distal relief to expose the bone and preparation of a minimal mucoperiosteal flap was made (Figure 5).” why was it made?

-> Thank you for this aspect. Instead of using flapless implantation, we preferred a minimal mucperiosteal flap technique in order to get an intraoperative overview and to assess the bone quality. line 158, 160

is the connection of the temporary restoration shorter than the implant connection?

-> Thank you for your question. The connection of the temporary restoration has the same geometry as the definitive implant connection.

“with default settings for PMMA-based high-density polymer blocks (Telio CAD, Ivoclar vivadent, Schaan, Liechtenstein) (Figure 12) and fabricated utilizing a five-axis milling unit (imes icore 350i, Eiterfeld, Germany).” how was the screw channel obtained? What kind of link?

-> Thank you for this aspect. The geometry of the TRI Implant restorations has been stored by the manufacturer in our exocad database. The five-axis milling unit enables the screw channel to be milled, so blanks without a prefabricated hole/screw channel can be used. On the other hand, there is, for example, a (chairside) milling unit with less than five axes, which requires a prefabricated block with screw channel. I have included this aspect in line 192-196.

In the discussion:

“In the past, implants were restored using metal abutments and porcelain fused to metal crowns. Using a metallic component has numerous disadvantages. In addition to the high material and processing costs, the aesthetics in particular are compromised.” Why? if the implant components are shown, the failure is surgical and often non-prosthetic. I do not consider this consideration legitimate.

 -> Thank you for your point of view. I thought about your comment and you’re right. The aesthetics aspect is mostly a surgical issue, so I deleted that aspect of aesthetics. line 236-239.

Kind regards,

F.Hölken

Reviewer 2 Report

The assessed work presents the use of the CAD / CAM digital technique in implant treatment. Admittedly, this technique has been used for many years. An interesting clinical case is discussed in the paper.

The quality of the work is very good.

The description of the method used is correct.

Very good quality of photographic documentation. Discussion and conclusion clearly and correctly presented.

I have a question? Did the authors obtain the approval of the local ethics committee for this type of research?

Author Response

Reviewer 2:

Dear Reviewer 2,

Thank you very much for taking the time to assess our manuscript and thank you very much for the positive feedback.

Thank you for your question: “did the authors obtain the approval of the local ethics committee for this type of research?”

As this is only a case report and not a clinical trial, no ethics committee was involved. The implant system is certified for medical device and FDA approved. It’s available on the market for any dentist. The patient agreed in the letter of consent to the documentation, photographs, and the publication of this manuscript. She does not receive any disadvantage as a result of this treatment.

Kind regards,

Hölken